# Resilience of *Aspergillus westerdijkiae* Strains to Interacting Climate-Related Abiotic Factors: Effects on Growth and Ochratoxin A Production on Coffee-Based Medium and in Stored Coffee

**DOI:** 10.3390/microorganisms8091268

**Published:** 2020-08-20

**Authors:** Asya Akbar, Angel Medina, Naresh Magan

**Affiliations:** Applied Mycology Group, School of Water, Energy and Environment, Cranfield University, Cranfield, Bedfordshire MK43 0AL, UK; asyaakbar@live.com (A.A.); a.medinavaya@cranfield.ac.uk (A.M.)

**Keywords:** *Aspergillus*, ochratoxin A, climate change, growth, carbon dioxide, temperature, water stress

## Abstract

We examined the resilience of strains of *Aspergillus westerdijkiae* in terms of growth and ochratoxin A (OTA) production in relation to: (a) two-way interacting climate-related abiotic factors of water activity (a_w_, 0.99–0.90) × temperature (25–37 °C) on green coffee and roasted coffee-based media; (b) three-way climate-related abiotic factors (temperature, 30 vs. 35 °C; water stress, 0.98–0.90 a_w_; CO_2_, 400 vs. 1000 ppm) on growth and OTA production on a 6% green coffee extract-based matrix; and (c) the effect of three-way climate-related abiotic factors on OTA production in stored green coffee beans. Four strains of *A. westerdijkiae* grew equally well on green or roasted coffee-based media with optimum 0.98 a_w_ and 25–30 °C. Growth was significantly slower on roasted than green coffee-based media at 35 °C, regardless of a_w_ level. Interestingly, on green coffee-based media OTA production was optimum at 0.98–0.95 a_w_ and 30 °C. However, on roasted coffee-based media very little OTA was produced. Three-way climate-related abiotic factors were examined on two of these strains. These interacting factors significantly reduced growth of the *A. westerdijkiae* strains, especially at 35 °C × 1000 ppm CO_2_ and all a_w_ levels when compared to 30 °C. At 35 °C × 1000 ppm CO_2_ there was some stimulation of OTA production by the two *A. westerdijkiae* strains, especially under water stress. In stored green coffee beans optimum OTA was produced at 0.95–0.97 a_w_/30 °C. In elevated CO_2_ and 35 °C, OTA production was stimulated at 0.95–0.90 a_w_.

## 1. Introduction

Coffee is an important economic export crop for many lower middle income countries (LMICs) in both South America, South East Asia and Africa. The production of both Arabica and Robusta coffee involves harvesting of the ripe cherries, fermentation and subsequent drying steps. During the latter phase, uneven drying can allow colonization by mycotoxigenic fungi and contamination with mycotoxins, especially ochratoxin A (OTA). This is predominantly due to the presence of species from the *Aspergillus* section *Circumdati* group, especially *Aspergillus westerdijkiae*. The other two important species in this group are *Aspergillus ochraceus* and *Aspergillus steynii* [1]. Other species that have been suggested to possibly contaminate coffee as well as other commodities such as cocoa and grapes are members of the *Aspergillus* section *Nigri* group (e.g., *Aspergillus carbonarius*). There are thus legislative limits for the maximum amount of OTA in both green and roasted coffee in many countries including the EU [2].

At the present time there has been significant interest in the impact that climate-related abiotic factors may have on the colonization of economically important commodities with mycotoxgenic fungi and mycotoxins [3,4]. It has been shown that changes in temperature (4–5 °C), increased atmospheric CO_2_ (from 400 to 1000 ppm) and drought/flooding episodes would have significant impacts on agronomy and security and safety of such commodities [3,5,6]. There is little knowledge of the impact of such interacting climate-related abiotic conditions on the functioning of key ochratoxigenic fungi or on the capacity for OTA production both in vitro and in situ [7]. Certainly, the biodiversity of microbial communities will change during plant growth, and this may change the dominance of different microbial communities on such commodities when entering the drying and storage phases [8]. This also applies to coffee beans where contamination mainly occurs either prior to harvest as the cherries ripen or contamination from soil during drying after the fermentation process. Inoculum mainly resides in soil, and it has been shown that ochratoxigenic fungi such as *A. ochraceus* (=*A. westerdijkiae*) and *Penicillium verrucosum* are very tolerant of soil matric water potential stress, and are thus active in soil and able to effectively colonize crop debris, allowing the subsequent contamination of drying coffee beans and other commodities [9,10].

Recent studies with mycotoxigenic species such as *Aspergillus flavus* have shown that while the colonization of staple food grains such as maize has been relatively unaffected by interacting CC-related abiotic factors (2–4 °C, 400 vs. 1000 ppm CO_2_ and drought stress) there has been significant impact on aflatoxin B_1_ contamination [3,4,11]. Some studies have also suggested that there may be differences in the effects of such interacting abiotic factors on OTA production by *A. westerdijkiae* and *A. carbonarius* on coffee-based media [7]. Studies with *Fusarium verticillioides* and fumonisin production on maize between silking and maturity showed that the biomass of the pathogen increased but that toxin production was unaffected by use of elevated CO_2_ exposure. However, exposure of maize plants to drought stress × elevated CO_2_ exposure stimulated contamination of ripening maize cobs with fumonisins [12,13]. There is thus interest in the resilience of strains of *A. westerdijkiae*, which is considered to be mainly responsible for the OTA contamination of coffee under interacting climate-related abiotic factors.

The objectives of this study were thus to examine: (a) the effect of in vitro effects of two-way abiotic stress factors of a_w_ × temperature on four strains of *A. westerdijkiae* (B 2, CBS 121986, 2A3, C1/1); (b) the effect of three-way interacting climate-related abiotic factors (drought stress (a_w_) × temperature (4 °C above optimum × existing and elevated CO_2_) on in vitro growth and OTA production on coffee-based matrices by two strains (B 2, CBS 121986); and (c) to determine the in situ effects of these three-way climate-related abiotic factors on OTA contamination of stored raw coffee beans by these two strains of *A. westerdijkiae*.

## 2. Materials and Methods

### 2.1. Fungal Isolate and Spores Preparation

The strains used were all from the *Aspergillus* section *Circumdati* group. Initially, four strains of *A. westerdijkiae* (B 2, CBS 121986, 2A3, C1/1), all isolated from green coffee beans, were used in this study. These were all confirmed producers of ochratoxin A using coconut cream agar and a conducive Yeast Extract Sucrose (YES) medium and quantifying production after 10 days growth.

### 2.2. Coffee-Based Agar Media Preparation

Agar media were initially prepared from both green and roasted Arabica coffee beans to examine the effect of coffee extract type and environmental interacting factors on growth. This involved milling either 300 g of green Arabica or roasted Arabica coffee beans. These were placed into a flask containing 1 L of water and boiled for 30 min to obtain concentrated coffee extracts. A double layer of muslin was used to filter the resulting concentrated mixtures. These were used at 6% concentration and modified with glycerol/water solutions to obtain a_w_ values of 0.99, 0.98, 0.95 and 0.90. A 2% technical agar (Thermo Fisher Scientific Oxoid Ltd., Basingstoke, Hampshire, UK) was added for solidifying the media prior to autoclaving at 121 °C for 15 min. The molten cooled agar media (green coffee extract agar (GCEA) and roasted coffee extract agar (RCEA)) were mixed and carefully poured into 9-cm Petri plates (approx. 17.5 mL per plate) in a sterile flow bench and allowed to cool. They were then stored in separate plastic bags at 4 °C until being returned to ambient conditions for inoculation. The final a_w_ levels were checked with a Aqualab 4TE a_w_ measuring meter (Decagon Devices, Inc., Pullman, WA, USA) and found to be within 0.003 a_w_ of the target level.

### 2.3. Inoculation for In Vitro Studies on Coffee-Based Media

The inoculum was prepared by growing each strain on the 6% GCEA at 30 °C for seven days. Spore suspensions were prepared by agitating the colony surface with a sterile spatula in 9 mL of sterile distilled water containing 0.05% Tween 80. All treatments and replicates in the experiments were centrally point-inoculated with 7–10 μL of the spore suspensions.

### 2.4. Comparison of the In Vitro Resilience in Relation to a_w_ × Temperature Stress on Growth of Strains of A. westerdijkiae on Green and Roasted Coffee-Based Agar Media

The GCEA and RCEA a_w_ treatments were inoculated and incubated at 25, 30 and 35 °C for 10 days. Growth of the colonies was measured in two directions at right angles to each other every two days. The colony radius (mm) vs. time (days) for each replicate of each strain under the different treatment conditions was plotted in Microsoft Excel. After data plotting, a linear model was used to calculate the relative growth rates (mm/day). The growth rates were obtained as the slope of the line. The square of the linear correlation coefficients was ≥0.98. The square of the linear correlation coefficients was ≥0.98. At the end of the experiment the ochratoxin A production was quantified [7]. These studies were carried out with all four strains.

### 2.5. In Vitro Effect of Climate Change Related Interacting Abiotic Factors on Growth and Ochratoxin A Production by the A. westerdijkiae Strains

The experiments were conducted using only the 6% GCEA as detailed previously with two strains of *A. westerdijkiae* (B 2, CBS 121986). The GCEA media were adjusted to 0.95, 0.98 and 0.99 a_w_, respectively. These were again autoclaved at 121 °C for 15 min, well-mixed, then approx. 17.5 mL was poured into 9-cm sterile Petri plates, cooled and kept at 4 °C until use. The final a_w_ levels were again checked with a water activity meter (Aqualab 4TE; Decagon Devices, Inc., Pullman, WA, USA).

Inoculated replicates and treatments were placed in plastic environmental chambers. Inoculated treatments of the same a_w_ were enclosed together in the environmental chambers containing switchable valves at each end: one for CO_2_ intake and the other for exit. Two 500-mL beakers of glycerol/water solution with the same a_w_ as the treatment were included in the chamber to maintain the same equilibrium relative humidity (ERH) as the media a_w_.

The chambers were flushed with either 5 L of air (400 ppm CO_2_) or 1000 ppm CO_2_ from a specialty gas cylinder (British Oxygen Company Ltd., Guilford, Surrey, UK; 1000 ppm CO_2_ cylinder) for about 10–15 min, then the valves were sealed as detailed previously [14]. The environmental chambers were flushed every 24 h. They were incubated for 10 days at 30 and 35 °C, and fungal growth was measured every two days. At the end of this period the cultures were removed for OTA analysis.

Growth of the colonies was measured as described previously every two days. Immediately after measurement the environmental chambers were flushed with the treatment CO_2_ for 10–15 min and then sealed and incubated at the treatment temperature.

### 2.6. In Situ Effect of Climate Change Abiotic Factors on Ochratoxin A Contamination of Stored Coffee Beans Inoculated with A. westerdijkiae and A. ochraceus Strains

The in situ effect of interacting climate change abiotic factors on OTA production by the *A. westerdijkiae* strains was determined from stored raw coffee beans. Initially, 15 kg of raw Arabica coffee beans were gamma-irradiated at 12–15 Kgys (Synergy Health, Swindon, Berks, UK). This removed any microbial contamination from the surface and internally in the coffee beans. Plating these beans on Nutrient, malt extract or DG18 media showed no growth of any colonies. The coffee beans were used to build a moisture adsorption curve, which was prepared by the addition of known amounts of water to 10 g of green coffee bean sub-samples and stored at 4 °C for 24 h to allow water adsorption. The samples were then removed and, after equilibration at 25 °C, the a_w_ of the hydrated coffee beans was measured using an Aqualab 4TE (Decagon Devices Inc., Pullman, WA, USA). The coffee bean samples were then dried at 110 °C for 24 h and kept in a desiccator at room temperature for 1 h, then weighed to determine the moisture content.

Subsequently 325 g of irradiated raw coffee was weighed and water was added using the water adsorption curve to obtain the required target experimental a_w_ levels of 0.97, 0.95 and 0.90 and kept at 4 °C for 24 h for equilibration with occasional shaking. They were then returned to laboratory temperature and allowed to equilibrate. The coffee beans were then divided into six sub-samples (50 g) in sterile solid substrate culture vessels (Magenta, Sigma-Aldrich Ltd., St. Louis, MO, USA) with permeable microporous membrane lids in a sterile flow bench.

Inoculation of coffee treatments was done for the each of the *A. westerdijkiae* strains (B 2, CBS 121986). The spore inoculum was obtained from cultures grown on 6% GCEA at 25 °C for seven days and spore suspensions prepared as detailed previously. The concentration was adjusted by dilution to approx. 10^4^ spores/mL by using a hemocytometer. Using the methodology of Palacios-Cabrera et al. [15], 0.5 mL of spore suspension (10^4^ CFUs/mL) of each strain was added to the 50 g of raw green coffee beans and shaken well. Twenty-five grams of coffee beans were used as a control at each a_w_ level. The replicates of the same treatment were placed in the environmental chambers. The methodology was the same as that used for the in vitro studies for CO_2_ flushing of either air (400 ppm) or 1000 ppm CO_2_. The inoculated coffee beans were incubated for 12 days at 30 and 35 °C. At the end of the storage period, the samples were all dried at 50 °C in a drying oven and then 25 g were ground for OTA analyses. These were stored at −20 °C until OTA extraction and quantification.

### 2.7. Ochratoxin A Extraction and Quantification

#### 2.7.1. In Vitro Studies

Five plugs (5 mm) were taken across the colonies with a surface-sterilized cork-borer and transferred to 2-mL Eppendorf tubes and weighed. To each Eppendorf tube, 1000 µL methanol was added. The samples were then shaken using a KS 501 digital orbital shaker for 30 min and centrifuged for 10 min at 15,000× *g*. The supernatant was filtered and analyzed with the HPLC system (Agilent, Berks, UK). Twenty μL of the extracted toxin from the treatments and replicates were injected into the HPLC system. The conditions for OTA detection and quantification were as follows:Mobile Phase: Acetonitrile (57%), water (41%), acetic acid (2%)Column: 120CC-C18 column (Poroshell 120, length 100 mm, diameter 4.6 mm, particle size 2.7 micron; 600 Bar)Temperature of column: 25 °CExcitation: 330 nmEmission: 460 nmFlow rate: 1 mL min^−l^Volume of sample injected: 20 µLRetention time: Approx. 2.49 minRun time: 17 minLimit of detection: 0.01 ng g^−1^Limit of Quantification: 0.039 ng g^−1^

#### 2.7.2. In Situ Ochratoxin A Quantification

For the in situ samples, initial clean-up was done using Neogen immunoaffinity columns (Neogen, Neocolumn method). Ten grams of milled of dried coffee beans were extracted with a 50 mL methanol/water (70:30) solution in 1% sodium bicarbonate. The extracts were then filtered, and 5 mL were diluted with 45 mL phosphate buffered saline (PBS/Tween (0.01% v/v) and applied to an immunoaffinity column (Neogen Europe Ltd., Auchincruive, Ayr, Scotland, UK). Of this mixture, 1.5 mL was dried and 0.5 mL of acetonitrile/water (50:50) was added. The final extracts were analyzed by HPLC as detailed previously. The retention time of OTA under the conditions described was approximately 2.5 min. The mobile phases used were acetonitrile (57%), acetic acid (2%) and water (41%) [16].

### 2.8. Statistical Analyses

A full factorial design was applied for the two-way (4 × 5) factor experiments: water activity and temperature. For each treatment, a water activity × temperature combination was carried out in triplicate for growth rate and OTA production, and the experiment repeated once.

Normality was checked using a Kolmogorov–Smirnov test. Analysis of data including factors, responses and their interactions were examined by a Kruskal–Wallis (non-parametric) test if the data were not normally distributed. For normally distributed data, the datasets were analyzed using a Minitab 16 package (Minitab Inc., 2010. State College, PA, USA). The statistical significance level was set at *p* ≤ 0.05 for all single and interacting treatment factors.

For climate-change-related factors, a full factorial design with three factors (water activity, temperature and CO_2_) was applied. Each treatment, a_w_ × temperature × CO_2_ combination was carried out in triplicate, both for growth rate assessment and OTA production, then repeated once. Normality was checked using a Kolmogorov–Smirnov test. Analysis of data and the effects of a_w_, temperature, CO_2_ and their interactions were examined by a Kruskal–Wallis (non-parametric) test if the data were not normally distributed. For normally distributed data, the datasets were analyzed using the Minitab 16 package (Minitab Inc., 2010. State College, PA, USA). The statistically significant level was set at *p* ≤ 0.05.

## 3. Results

### 3.1. Comparison of Growth and Ochratoxin A Production by Strains of A. westerdijkiae on Green and Roasted Coffee-Based Media

Figure 1 and Figure 2 show the relative growth rates of the four strains of *A. westerdijkiae* in relation to two-way interacting abiotic factors of a_w_ × temperature on coffee media made from both green and roasted coffee extracts. Overall there appeared to be slightly lower growth rates on the roasted than the green coffee-based media. Optimum growth was predominantly at 25 °C and 0.98 a_w_ for all the strains on both nutritional media. All the four strains grew over the a_w_ range 0.95–0.99 on both media. However, growth was very slow at 0.90 a_w_ when compared to the other a_w_ levels examined. Indeed, the CBS 121986 strain was unable to grow at 35 °C and 0.90 a_w_.

Table 1 compares the actual OTA production by the four different strains of *A. westerdijkiae* in relation to growth under different a_w_ × temperature conditions on the two nutritional green and roasted coffee-based media. This shows that there was very little if any OTA production on the RCEA by any of the strains. In contrast, on the GCEA both B 2 and CBS 121896 strains produced the highest amounts of OTA with an optimum at 0.98 a_w_ and 30 °C. Lower amounts were produced at 0.95 a_w_ and 30 °C.

Table 2 summarizes the statistical analyses for mycelial growth and OTA production for the four strains of *A. westerdijkiae* at different a_w_ levels on the two media types and different temperature conditions. The growth rate was significantly (*p* < 0.05) affected by a_w_ (0.90, 0.95, 0.98 and 0.99) and temperature (25, 30 and 35 °C) for all strains. There was no significant difference between the other three strains examined (2A3, B 2, CBS 121986).

For OTA production, a_w_ was a significant factor for all strains. There were similar amounts of OTA produced by *A. westerdijkiae* (C1/1) in the two media, although for the other *A. westerdijkiae* strains (2A3, B 2, CBS 121 986) there was a significant effect of medium type. Overall, the strains grew best at 25 °C in both GCEA and RCEA and produced most OTA at 30 °C. The individual factors assayed (water activity, temperature, coffee medium type) as well as the interaction of a_w_
*×* temperature *×* coffee media (GCEA; RCEA) had a significant influence on growth and OTA production. Based on these results, only two strains (B 2, CBS 121986) were subsequently used to examine the effects of climate-related abiotic factors on growth and OTA production.

### 3.2. In Vitro Effect of Interacting Climate-Related Abiotic Factors of Water Activity × Elevated CO_2_ × Temperature on Growth and OTA Production by Strains of A. westerdijkiae and A. ochraceus

#### 3.2.1. In Vitro Effects of Interacting Climate-Related Abiotic Factors on Growth

Figure 3 and Figure 4 compare the growth rate for the two strains of *A. westerdijkiae* when exposed to air and elevated CO_2_ at different a_w_ levels at 30 and 35 °C, respectively. At 30 °C optimum growth was at 0.98 a_w_ then 0.99 a_w_, with lower growth rates at 0.95 a_w_. However, when temperature was elevated to 35 °C, growth was significantly influenced by both a_w_ and CO_2_ (1000 ppm) and their interaction when compared to that at 30 °C. The two *A. westerdijkiae* strains grew much more slowly at 35 °C, with the fastest growth at 0.95 a_w_. Much less growth occurred at 0.98 a_w_, and there was practically no growth at 0.99 a_w_ in air or with elevated CO_2_.

Statistically, the factors assayed (water activity and CO_2_), as well as their interactions, had a significant effect on mycelial growth for the strains examined at 30 and 35 °C using the Kruskal–Wallis and ANOVA tests (Table 3). Growth was significantly (*p* < 0.05) affected by elevated CO_2_ at 30 °C and 0.95 and 0.99 a_w_ for the two *A. westerdijkiae* strains at all a_w_ levels. At 35 °C, there was a statistically significant effect on growth for the strains. The effect of the various factors and their interactions were more significant at 35 °C than at 30 °C. Changing temperature from 30 to 35 °C had a significant effect on growth.

#### 3.2.2. In Vitro Effects of Climate-Related Interaction of Abiotic Conditions on OTA Production

Figure 3 and Figure 4 also show the effect of the three-way interacting climate-related factors on in vitro OTA production by the strains of *A. westerdijkiae*. There was little effect of elevated CO_2_ on OTA production at 30 °C. However, OTA production was reduced slightly when exposed to 1000 ppm CO_2_ especially at 0.97 a_w_.

When the temperature was increased to 35 °C in the presence of elevated CO_2_ (1000 ppm), there was a stimulation effect on OTA production at 0.98 a_w_ by the two *A. westerdijkiae* strains. Thus, for strain B 2 there was a significant stimulation under water stress at 0.90 a_w_, with OTA production increased from about 7.0 to 680 ng g^−1^ when exposed to 1000 ppm CO_2_.

Statistically, OTA production was significantly affected by elevated CO_2_, a_w_ and the interaction between a_w_ and elevated CO_2_ at 30 and 35 °C (Table 4).

### 3.3. In Situ Effect of Water Activity × Elevated CO_2_ × Temperature on OTA Production at 30 and 35 °C in Stored Coffee Beans by A. westerdijkiae Strains

Table 5 shows the effect of climate-related factors (30 vs. 35 °C; 400 vs. 1000 ppm CO_2_; 0.90, 0.95 and 0.97 a_w_) on OTA production by the strains of *A. westerdijkiae* examined. High amounts of OTA were produced by these two strains of *A. westerdijkiae* at 0.90 and 0.95 a_w_ in elevated CO_2_ (1000 ppm) at 30 °C. In some cases there was some stimulation by elevated CO_2_ at 30 °C, especially under water stress. This produced 4598.8 (B 2 strain) and 3974.2 ng g^−1^ (CBS121896) at elevated CO_2_ in contrast to much lower amounts in the normal 400 ppm CO_2_ treatment levels.

When the temperature was increased from 30 to 35 °C, OTA production for these two strains was reduced at different a_w_ levels and CO_2_ (1000 ppm). They both produced more OTA when exposed to elevated CO_2_ at 35 °C.

Statistically, Table 6 shows that the effect of interacting a_w_ × elevated CO_2_ (1000 ppm) × temperature on the strains. There was a significant increase (*p* = 0.05) in OTA production in the presence of elevated CO_2_ at 0.90 and 0.97 a_w_. A_w,_ CO_2_ and their interaction significantly (*p* < 0.05) affected OTA production by the B 2 strain in stored green coffee beans at 30 °C. There were no effects of a_w_ on the CBS type strain of *A. westerdijkiae* (CBS 121986).

## 4. Discussion

### 4.1. In Vitro Effects of Climate-Related Abiotic Factors on Growth and OTA Production by Strains of A. westerdijkiae

This study has examined the resilience of strains of *A. westerdijkiae* in relation to two-way interacting abiotic factors (a_w_ × temperature) and three-way interacting climate-related factors (a_w_ × temperature × CO_2_) on growth and OTA production in vitro on coffee extract-based matrices and stored coffee. Interestingly, while growth was similar on both green coffee extract and roasted coffee extract media, OTA production on the roasted coffee-based medium was practically undetectable. Certainly the strains were able to grow well over a wide range of temperatures, with reduced growth at 0.90 a_w_. This suggests that rapid colonization of drying green coffee would occur in the range of 25–35 °C under 0.90–0.98 a_w_ conditions and result in a higher risk of OTA contamination. The two strains chosen for further studies did produce significant amounts of OTA, and were thus used for more detailed studies on the impact of three-way interacting climate-related abiotic factors.

Previous studies by Taniwaki et al. [17,18] reported that *A. ochraceus* (=*A. westerdijkiae*) was primarily responsible for the OTA contamination of green coffee, suggesting a minimum a_w_ for growth of about 0.85. Pardo et al. [19] suggested that in vitro growth on green coffee-based media was optimal for *A. ochraceus* (=*A. westerdijkiae*) strains at 0.95–0.99 a_w_ and 20–30 °C. They suggested the a_w_ minimum for germination was 0.80, and for mycelial growth was 0.85 a_w_. Maximum OTA production for *A. ochraceus* occurred at 0.99 a_w_ with no toxin produced at 0.80 a_w._ Previous studies suggest that optimum OTA production by *A. ochraceus* (=*A. westerdijkiae*) was at 0.95 a_w_, with no production at 0.90 a_w_. The maximum growth for the type strains of *A. westerdijkiae* on YES media occurred at 0.95 a_w_ and 30 °C. Optimum OTA production was at 25 °C [1]. Akbar et al. [13] used a 20% coffee-extract medium that influenced relative growth rates and OTA production although optimum conditions, especially for growth, were relatively similar. With the exception of the work of Pardo et al. [19,20] and Akbar et al. [13], most of the other studies were carried out have used defined laboratory media instead of a heterogenous coffee-based medium. The latter medium may provide more useful data and may more accurately simulate what may occur in situ.

This study has shown the resilience of the strains of *A. westerdijkiae* to two-way interacting climatic conditions that form a platform for examining in more detail the effects of three-way interacting climate-related abiotic factors by including exposure to existing and elevated CO_2_ levels (400 vs. 1000 ppm) [3]. Coffee beans during the drying and storage phase may be exposed to these three-way interacting conditions, which will influence both the dominance of toxigenic fungal groups including those from the *Aspergillus* section *Circumdati* and the section *Nigri* species [13,21,22]. The present study has focused on one of the key species from the section *Circumdati* only. Studies have shown, however, that slightly elevated CO_2_ levels combined with drought stress and increased temperature may enhance OTA production by strains of *A. westerdijkiae* on green coffee-based media. Interestingly, the impact on OTA production appeared to be more pronounced than the effects on colonization rates. In addition, elevated CO_2_ (1000 ppm) + elevated temperature (35 °C) increased OTA production when compared with 30 °C for one of the *A. westerdijkiae* strains (B 2). Both strains had optimum growth at 0.95 a_w_ and 35 °C, while at 30 °C, the optimum was at 0.98 a_w_.

Most previous studies that have examined the impact of a_w_ × elevated CO_2_ have focused on modified atmosphere storage to try and control OTA production post-harvest in both coffee and other commodities. Cairns-Fuller et al. [23] suggested that 50% CO_2_ was required to inhibit growth and OTA production by *P. verrucosum* in moist grain by >75% at 0.90–0.995 a_w._ Paster et al. [24] reported that no growth of *A. ochraceus* (=*A. westerdijkiae*) occurred at 80 or 100% CO_2_, and that growth was partially inhibited by 60% CO_2_. Similarly, Pateraki et al. [25] found that 50% CO_2_ inhibited *A. carbonarius* growth after five days. Some studies have reported a reduction in growth rate at 5–10% of CO_2_ for some species at low a_w_ levels, including an increase in the lag phases prior to growth. Although at ≥15% CO_2_ most strains showed growth inhibition, especially under water stress [26]. Valero et al. [27] found that there was a significant reduction in growth and OTA production by *Aspergillus* section *Nigri* species such as *A. carbonarius* and *A. niger* at 1% O_2_ when combined with 15% CO_2_.

Medina et al. [3] suggested that often while exposure of mycotoxigenic fungi such as *A. flavus* to interacting climate-related abiotic factors has little effect on growth, this does however result in a significant stimulation of aflatoxin B_1_ (AFB_1_) production on milled maize-based media and in stored maize. Indeed, recent studies have shown that the kinetics of AFB_1_ production change over time when this species is exposed to climate-related abiotic factors [28]. This is supported by relative expression of key structural and regulatory genes involved in the biosynthesis of aflatoxins. Mycotoxigenic fungi show significant plasticity in physiologically responding to stress factors, and thus three-way interacting stress factors may have a more significant impact on the colonization of different commodities and production of mycotoxins and other secondary metabolites by these toxigenic species.

### 4.2. In Situ Effect of Three-Way Interacting Climate-Related Abiotic Factors on OTA Production in Stored Green Coffee

This study examined the effects of three-way climate-related abiotic factors on contamination of stored green Arabica coffee beans with OTA when inoculated with individual strains of ochratoxigenic *Aspergillus* section *Circumdati* species. Based on the water adsorption curve, the range of a_w_ levels used represent a moisture content (m.c.) of between approx. 40% and 20–22%, which is equivalent to the range of 0.90–0.97 a_w_. This study has shown that there was a high OTA contamination of stored green coffee when colonized by *A. westerdijkiae* strains, especially at 0.95 and 0.97 a_w_ and 30 °C, regardless of CO_2_ levels. At 35 °C, the OTA contamination levels were significantly lower than at 30 °C regardless of other abiotic treatments. However, at 35 °C there was higher OTA contamination levels in these two *A. westerdijkiae* strains at 0.95 and 0.90 a_w_ and 1000 ppm CO_2_. This suggests that the stress of interacting abiotic factors results in a stimulation of OTA production by this species.

Previously, Palacios-Cabrera et al. [15] found high amounts of OTA produced by *A. carbonarius* in irradiated raw coffee beans when stored at 100% Equilibrium Relative Humidity (=1.00 a_w_). Similarly, maximum growth rates of *A. ochraceus* on irradiated green coffee beans was found to occur at 30 °C and 0.95–0.99 a_w_, which was similar to the in vitro studies in the present work. Other studies on irradiated coffee beans have shown OTA production between 40–17,000 ng g^−1^ over this a_w_ range at 30 °C [20]. This study also showed that limited OTA was produced at 10 °C and 0.80 a_w_. It has been specified that during post-fermentation and drying, the final safe storage m.c. for green coffee beans should be in the range of 10–12% (approx. 0.65–0.70 a_w_ [29]). However, uneven drying can result in pockets of wetter coffee beans that are conducive to colonization by ochratoxigenic species.

Prior to the reclassification of the *Aspergillus* section *Circumdati*, Taniwaki et al. [18] found that there was very low accumulation of OTA by *A. ochraceus* (=*A. westerdijkiae*) at 0.80 a_w_ and 25 °C, with an increased contamination at 0.95 a_w_ after three weeks storage. Bucheli et al. [22] also found the lowest a_w_ levels for OTA contamination of green coffee by *A. ochraceus* (=*A. westerdijkiae*) were 0.85 and 20–30 °C. However, these earlier studies did not examine the effects of >30 °C on interactions with other abiotic factors, especially water stress and elevated CO_2_. Akbar et al. [7] found that there were differential effects of three-way interacting climate-related abiotic factors on species of the section *Circumdati* and section *Nigri*. The species from the latter group (*A. carbonarius*, *A. niger*) appeared to be less affected by such abiotic changes and thus had a better resilience to such conditions. This may of course have implications, such as the fungal community structure and dominance of different toxigenic species colonizing the ripening coffee cherries pre-harvest, and post-harvest drying and processing may change under climate-related conditions and influence both colonization and mycotoxin contamination in the coffee production chain [3,5].

In the in situ study only Arabica coffee beans were used. There are significant and inherent differences in the caffeine concentration between this type (0.6%) and Robusta (4%) [30]. Caffeine has been shown to have some anti-fungal inhibitory effects and can inhibit both growth and mycotoxin production significantly. For example, Akbar et al. [21] showed that 0.5–1.0% caffeine inhibited growth of toxigenic species from both Section *Circumdati* and *Nigri* groups on a conducive defined medium. Thus, it may be important to compare the effect of climate-related abiotic factors on the colonization of both Arabica and Robusta coffee beans to evaluate and compare the impacts on OTA production. Inherent caffeine concentrations may inhibit colonization by some toxigenic species and perhaps influence the levels of OTA contamination that may occur. Robusta coffee beans are mainly grown at lower altitudes, and may have better tolerance to climate-related abiotic factors than Arabica, which is grown at higher altitudes and may be more sensitive to climate-related agronomic factors [31]. In addition, changes in diversity of pests could lead to increased damage to ripening coffee beans, and such damage has been shown to have an impact on infection by toxigenic fungi and perhaps also toxin contamination levels post-harvest [32].

Other important abiotic factors have not been considered such as UV radiation and fluctuations in temperature × water availability and exposure to elevated CO_2_ conditions [32,33]. Some of these factors have been shown to impact growth and OTA production by strains of *A. carbonarius* isolated from grapes [33]. The impact of acclimatization of strains of species from both sections *Circumdati* and *Niger* have not been addressed. Studies by Vary et al. [34] on *Fusarium graminearum* and by Medina et al. [3] on *A. flavus* have shown that exposing these species to elevated CO_2_ conditions for 10–20 generations can result in enhanced tolerance and pathogenicity and lead to increased mycotoxin production.

## 5. Conclusions

This study has shown that in terms of colonization of both green and roasted coffee-based matrices strains of *A. westerdijkiae* have very good resilience to two-way climate-related interacting abiotic factors. However, on media made from roasted coffee beans there was very little OTA contamination when compared to that of green coffee-based media. Three-way interacting abiotic climate related factors had some effect on growth of the strains of *A. westerdijkiae*, especially at 35 °C and 0.98–0.95 a_w_ with elevated CO_2_. However, there was significant stimulation of OTA production. The data suggest that in stored green coffee beans, *A. westerdijkiae* is able to produce OTA when temperature is increased to 35 °C + 1000 ppm CO_2_ and drought stress is imposed. The present studies used gamma-irradiated coffee beans that had no competition from the resident natural surface mycobiota. Studies are needed to examine how such interactions may affect the resilience of ochratoxigenic fungi and the contamination of different coffee cultivars with OTA. However, the type of data presented in this study is essential for the development of more accurate models to determine the relative risk of OTA in the coffee production and processing chain.

## Figures and Tables

**Figure 1 microorganisms-08-01268-f001:**
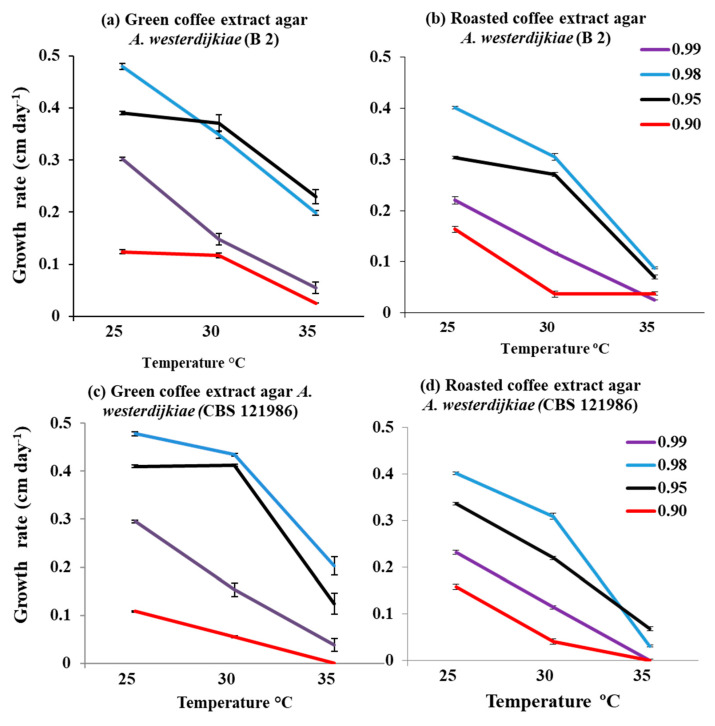
Effect of (**a**,**c**) green milled coffee extract (GCEA) and (**b**,**d**) roasted coffee extract (RCEA) media on growth (cm day^−1^) of two of the *A. westerdijkiae* strains in relation to different temperatures (25, 30, 35 °C) and water activity (a_w_) levels (0.99, 0.98, 0.95, 0.90) after 12 days colonization. Bars indicate SEM.

**Figure 2 microorganisms-08-01268-f002:**
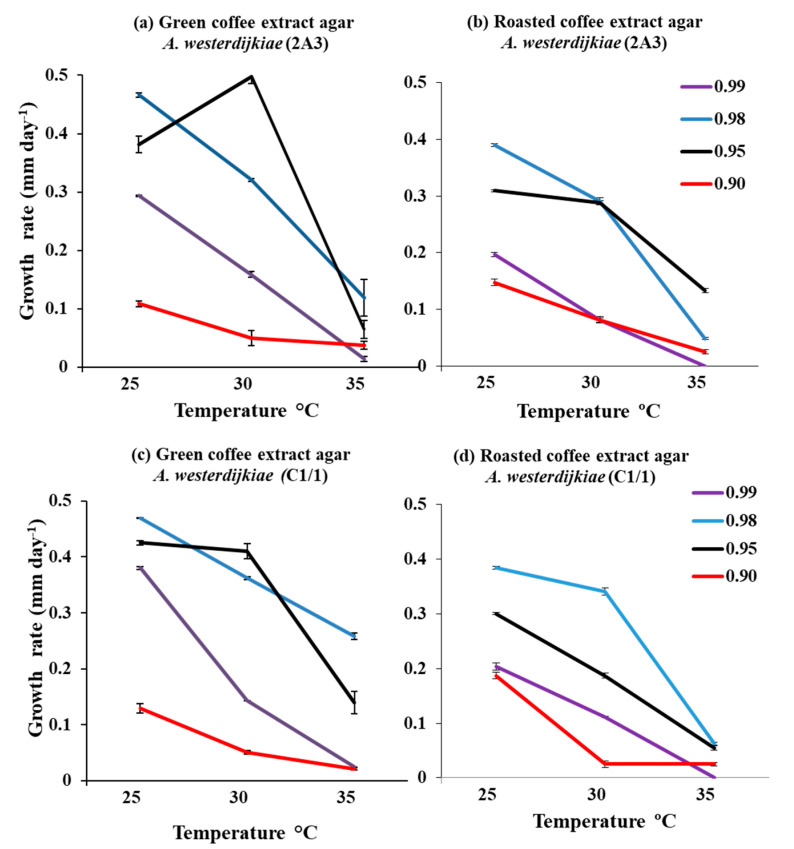
Effect of (**a**,**c**) green milled coffee extract (GCEA) and (**b**,**d**) roasted coffee extract (RCEA) media on growth (cm day^−1^) of the other two *A. westerdijkiae* strains in relation to different temperatures (25, 30, 35 °C) and water activity (a_w_) levels (0.99, 0.98, 0.95, 0.90) after 12 days colonization. Bars indicate SEM.

**Figure 3 microorganisms-08-01268-f003:**
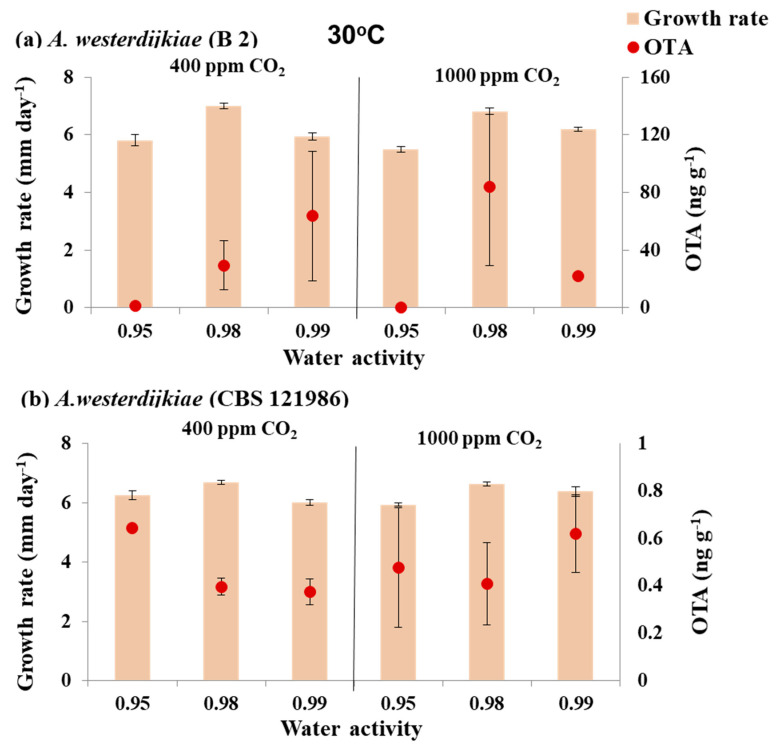
Effect of water activity × CO_2_ exposure at 30 °C on growth (mm day^−1^) and ochratoxin A (ng g^−1^) production by strains of *A. westerdijkiae* (**a**,**b**) on a 6% green coffee extract agar after 10 days incubation. Bars represent SEM. Please note that *Z*-axis has different ranges for OTA production.

**Figure 4 microorganisms-08-01268-f004:**
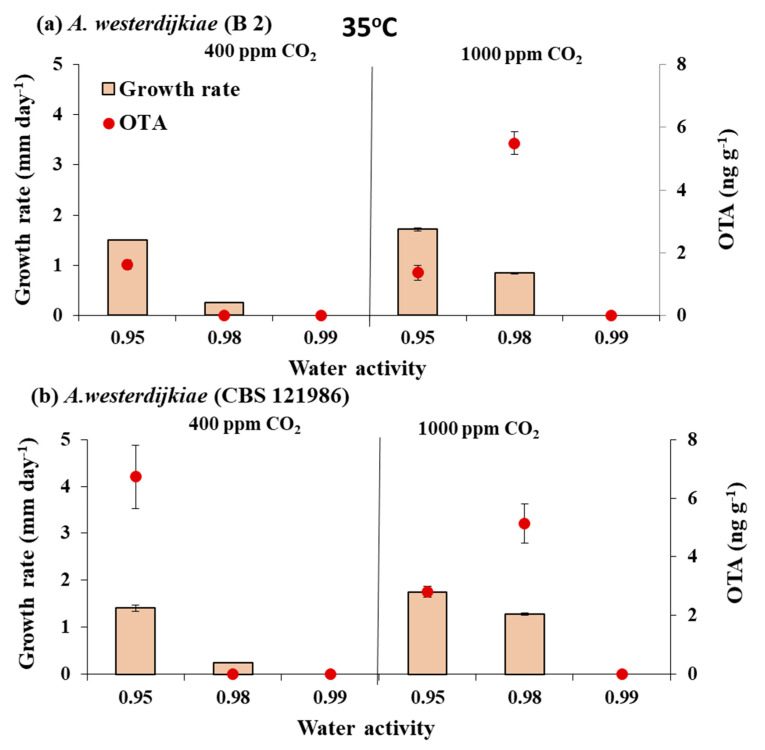
Effect of water activity × CO_2_ exposure at 35 °C on growth (mm day^−1^) and ochratoxin A (ng g^−1^) production by strains of *A. westerdijkiae* (**a**,**b**) on a 6% green coffee extract agar after 10 days incubation. Bars represent SEM.

**Table 1 microorganisms-08-01268-t001:** Effect of a_w_, temperature and type of coffee extract medium on ochratoxin A (OTA) production (ng g^−1^) by the four strains of *A. westerdijkiae* examined.

Temperature (°C)		25		30		35	
Strains/Medium/a_w_		GCEA	RCEA	GCEA	RCEA	GCEA	RCEA
*A. westerdijkiae* (2A3)	0.99	ND	ND	ND	ND	ND	ND
	0.98	4.1 + 1.1.	<LOQ	8.3 + 3.9	<LOQ	10.3	ND
	0.95	8.7 + 0.9	2.6 + 0.2	<LOQ	<LOQ	46.4	ND
	0.90	ND	<LOQ	ND	ND	ND	ND
*A. westerdijkiae* (C1/1)	0.99	<LOQ	ND	ND	ND	ND	ND
	0.98	<LOQ	<LOQ	48.3 + 13.8	ND	9.7	ND
	0.95	32.4 + 0.4	<LOQ	ND	ND	ND	ND
	0.90	ND	<LOQ	ND	ND	ND	ND
*A. westerdijkiae* (B 2)	0.99	ND	ND	40.3 + 5.1	ND	ND	ND
	0.98	9.4 + 5.2	<LOQ	1802.8 + 416	ND	5206.3 + 433	ND
	0.95	3.7 + 0.2	<LOQ	3.2 + 0.4	<LOQ	28.1 + 1.1	<LOQ
	0.90	ND	ND	<LOQ	ND	<LOQ	ND
*A. westerdijkiae* (CBS 121986)	0.99	<LOQ	ND	<LOQ	<LOQ	ND	ND
	0.98	<LOQ	<LOQ	280.4 + 17.2	ND	64.3 + 15.8	ND
	0.95	30.6 + 6.2	<LOQ	5.4 + 3.0	ND	52.81	ND
	0.90	ND	<LOQ	<LOQ	ND	ND	ND

Key: GCEA, green coffee extract agar; RCEA, roasted coffee extract agar; <LOQ, below Limit of Quantification; ND, no toxin detected.

**Table 2 microorganisms-08-01268-t002:** Summary of the statistical analyses of the effect of the factors on growth rate and OTA production by the four strains of *A. westerdijkiae* in relation to a_w_, temperature, coffee medium and their interactions using the Kruskal–Wallis test (non-normality data).

Strain	Temperature	Media Type	Water Activity	Response
*A. westerdijkiae* (2A3)	S	NS	S	Growth
	S	S	S	OTA (ng g^−1^)
*A. westerdijkiae* (C1/1)	S	S	S	Growth
	S	NS	S	OTA (ng g^−1^)
*A. westerdijkiae* (B 2)	S	NS	S	Growth
	S	S	S	OTA (ng g^−1^)
*A. westerdijkiae* (CBS 121986)	S	NS	S	Growth
	S	S	S	OTA (ng g^−1^)

Significance based on *p* ≤ 0.05; NS, not significant; S, significant.

**Table 3 microorganisms-08-01268-t003:** Summary of statistical effects on growth rate of the strains of *A. westerdijkiae* in relation to CO_2_, water activity (a_w_) and CO_2_ at 30 and 35 °C on a 6% green coffee extract medium using Kruskal–Wallis (non-normality data) and ANOVA (normality data) tests.

**Temperature 30 °C**				
**Strains**	**CO_2_ (1000 ppm)**	**Water Activity (a_w_)**	**CO_2_ × a_w_**	**Response**
*A. westerdijkiae* (B 2)	NS ^a^	S ^b^	NS ^a^	growth rate
*A. westerdikiae* (CBS 121986)	S ^a^	NS ^a^	-	growth rate
**Temperature 35 °C**				
*A. westerdijkiae* (B 2)	NS ^b^	S ^b^	NS ^a^	growth rate
*A. westerdijkiae* (CBS 121986)	NS ^a^	S ^a^	-	growth rate
**Temperatures 30 and 35 °C**				
**Strains**	**CO_2_ (1000 ppm)**	**a_w_**	**Temp (30 + 35 °C)**	**Response**
*A. westerdijkiae* (B 2)	NS ^a^	NS ^a^	S ^a^	growth rate
*A. westerdijkiae* (CBS 121986)	NS ^a^	S ^a^	S ^a^	growth rate

Key: ^a^, Kruskal–Wallis test; ^b^, ANOVA; NS, not significant; S, significant (*p* ≤ 0.05).

**Table 4 microorganisms-08-01268-t004:** Summary of statistical results for the OTA production by the strains of *A. westerdijkiae* in relation to a_w_, temperature and CO_2_ using Kruskal–Wallis (non-normality data) and ANOVA (normality data) tests.

**Strains**	**CO_2_ (1000 ppm)**	**a_w_**	**CO_2_ × a_w_**
*A. westerdijkiae* (B 2)	S ^a^	S ^a^	-
*A. westerdijkiae* (CBS 121986)	NS ^b^	S ^b^	NS ^b^
**Temperature 35 °C**			
*A. westerdijkiae* (B 2)	S ^a^	S ^a^	-
*A. westerdijkiae* (CBS 121986)	S ^a^	S ^a^	-
**Temperature 30 and 35 °C**			
**Strains**	**CO_2_ (1000 ppm)**	**a_w_**	**Temp 30 + 35**
*A. westerdijkiae* (B 2)	S ^a^	S ^a^	S ^a^
*A. westerdijkiae* (CBS 121986)	S ^a^	S ^a^	S ^b^

Key: ^a^, Kruskal–Wallis test; ^b^, ANOVA; NS, not significant; S, significant (*p* ≤ 0.05).

**Table 5 microorganisms-08-01268-t005:** Effect of interacting climate-related factors of water activity (a_w_), temperature (30 vs. 35 °C) and CO_2_ (400 vs. 1000 ppm) on the mean ochratoxin A contamination (ng g^−1^ ± SE) of green coffee beans stored for 10 days after inoculation with each of the strains.

Temperature (°C)		30		35	
CO_2_ Concentration (ppm)		400	1000	400	1000
Strains	a_w_				
*A. westerdijkiae* (B 2)	0.97	3976.9 ± 603.7	2760.4 ± 52.7	175.6 ± 0.2	14.4 ± 3.7
	0.95	4243.3 ± 571.4	4767.1 ± 372.1	128.8 ± 31.6	63.1 ± 21.3
	0.90	1644.3 ± 545.3	4598.9 ± 426.4	8.2 ± 0.2	680.2 ± 187.2
*A. westerdijkiae* (CBS 121986)	0.97	2681.3 ± 346.7	3395.5 ± 198.7	8.1 ± 0.4	12.7 ± 0.9
	0.95	2842.3 ± 325.1	3087.9 ± 225.4	8.2 ± 0.6	51.3 ± 28.6
	0.90	2679.3 ± 391.3	3974.2 ± 101.6	7.1 ± 0.7	69.5 ± 21.2

**Table 6 microorganisms-08-01268-t006:** Summary of statistical effect of climate-related abiotic factors on OTA production by the two strains of *A. westerdijkiae* in stored coffee beans examined in this study. Statistical analyses based on Kruskal–Wallis (non-normality data) and ANOVA (normality data) tests.

**Temperature (30 °C)**			
**Strains**	**CO_2_**	**a_w_**	**a_w_ × CO_2_**
*A. westerdijkiae* (B 2)	S ^b^	S ^b^	S ^b^
*A. westerdijkiae* (CBS 121986)	S ^b^	NS	NS
**Temperature (35 °C)**			
*A. westerdijkiae* (B 2)	S ^a^	S ^a^	N/A
*A. westerdijkiae* (CBS 121986)	S ^a^	NS ^a^	N/A
**Strains**	**CO_2_ (1000 ppm)**	**a_w_**	**Temp: 30 + 35 °C**
*A. westerdijkiae* (B 2)	NS ^a^	NS ^a^	S ^a^
*A. westerdijkiae* (CBS 121986)	S ^a^	NS ^a^	S ^a^

Key: S, significant (*p* < 0.05); NS, not significant (*p* > 0.05); ^a^, Kruskal–Wallis test; ^b^, ANOVA; N/A, not applicable.

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
