# Peer review of "Resilience of Aspergillus westerdijkiae Strains to Interacting Climate-Related Abiotic Factors: Effects on Growth and Ochratoxin A Production on Coffee-Based Medium and in Stored Coffee"

_microorganisms, 2020, doi:10.3390/microorganisms8091268_

Round 1

Reviewer 1 Report

The manuscript by Asya Akbar et al. describes the impact of climate-related abiotic factors (water activity, temperature, CO2) on the fungal growth and ochratoxin A production of different strains of Aspergillus westerdijkiae. Research conducted in in-vitro cultures on coffee media and in-situ on stored raw coffee beans is a part of a series of authors’ on the resilience of the Aspergillus strains to the different factors shaping their mycotoxigenic activity, very important in food production. Presented results deliver interesting results, although several points should be addressed before the manuscript is accepted for publication. 

Major remarks:

Manuscript suffers from much editorial weakness.

Lines 48-51: „Certainly, the biodiversity of microbial communities will change during plant growth and this may change the dominance of different microbial communities on such commodities when entering the drying and storage phases.” – please add the reference to this statement

The objectives: line 70: In point b the statement „effect of three-way interacting factors on...” is not clear.  Please clearly define which factors are taken into account.

Please clarify the taxonomic affiliation of tested fungal strains. In the text, the strains are defined as strains of A. ochraceus and in the tables, these same strains are present as an A. westerdijkiae. This same in case of subtitles of the manuscript and the descriptions of the tables, which contain both names, when the data in tables are characterised as a data of A. westerdijkiae. If they are treated by authors as synonyms, please use one species name.

Line 110: What decided about using only these two strains?

Figure 3. In all manuscript, data are presented in order from the highest to the lowest aw values. I suggest reorganizing of bars in Figure 3 according to this same order.

Minor remarks:

Line 14 and the rest of the manuscript: „…vs 35oC…” – please correct the degree symbol

 Line 18: „… 0.98 and …”– of what? Please add the unit

Line 19: „… green coffee-base dmedia…” – please correct

Line 72: „…A ocharceus.” – please correct

Line 86: „…autoclabng…” – please correct

Line 107: „Thes estudies…” – please correct

Line 139 and the rest of the manuscript: „aw” or „aw” ? – please use one form of the water activity symbol throughout the manuscript

Line 157: „…CO2…” – please correct

Line 160: „Mycotoxins extraction and analysis” - ?

Line 206: P-value equal to or less than

Line 209: „...mdia” – please correct

Line 219: „roatse…” – please correct

Line 221: please correct the degree symbols and „C”

Line 268: „…A. esterdijkiae…” – please insert „w”

Line 284: P-value equal to or less than

Figure 3b: y-axis description: „(mm day-1)” – please correct

Line 305: „…(a, b)on…” – please insert the space

Figure 4b: y-axis description: „(mm day-1)” – please correct

Lines 319, 320, 324, 329 in Table 3: – please correct the fungal species name

Lines 331 and 356:  P-value equal or less than

Lines 366, 369 in Table 5: „Asp.” – please correct the fungal species name

Line 405: „…froms …” – please correct

Line 415: „…strains…” – please remove italics

Line 444: „…moisture content (m.c.)…” – is this abbreviation (m.c.) used anywhere in the text? If not, I suggest to  remove it

Line 444: „…to 20-22%.,” – please remove the dot

Lines 501 and 503: „CO2” – please correct

References: - please correct the formatting of the references according to the rules of the journal

Reviewer 2 Report

The article deals with the‭ ‬interaction of different climate related abiotic factors‭ (‬temperature,‭ ‬water activity,‭ ‬CO2‭ ‬concentration‭) ‬on growth and ochratoxin A production of‭ ‬Aspergillus westerdijkiae‭ (‬former‭ ‬A.‭ ‬ochraceus‭) ‬on coffee-based medium and in stored coffee.‭

The experiments are well constructed‭ ‬and‭ ‬clearly presented,‭ ‬and‭ ‬they‭ ‬are understandable‭; ‬the experimental design was well structured and the conclusions are supported by data.

The Abstract is correct and provides an accessible summary of the paper. The Introduction section is a well-written part of this manuscript providing important literature on the background of this topic. The methodology and the discussion of the obtained results of the experiments is correct.

The main problem with the current state of this manuscript,‭ ‬that it is full with typing mistakes.
‭(The use of grammar check function of the word processor program would have useful.)

vs‭ – ‬correct:‭ ‬vs.
L14,‭ ‬L15,‭ ‬L59,‭ ‬L346

L19:‭ correct: ‬coffee-based media
L20:‭ correct: ‬roasted
L23:‭ ‬+‭ ‬or x,‭ ‬but consequently

L53:‭ correct: ‬Inoculum mainly
L54:‭ ‬vercuosum‭ ‬(sic‭!)‬ -‭ ‬correct:‭ ‬verrucosum‭
L54:‭ correct: ‬matrix

L60:‭ ‬delete: a „,‭” after [3 ] and „‬with have‭” ‬from the sentence
L62:‭ ‬Akbar et al.,‭ ‬2016 correct: [7]

L65:‭ ‬the use of‭ „‬CO2‭ ‬level‭” ‬would better

If‭ ‬Aspergillus westerdijkiae‭ (‬= A.‭ ‬ochraceus‭)‬,‭ ‬as it is written correctly at L53-L54‭ ‬and L390‭ ‬you can’t write,‭ ‬that you examined strains of‭ ‬Aspergillus westerdijkiae‭ ‬and‭ ‬A.‭ ‬ochraceus.
L72,‭ ‬L132,‭ ‬L149,‭ ‬L287,‭ ‬L312,‭ ‬L342,‭ ‬L374

L71:‭ correct: ‬three‭

L72:‭ ‬ocharceus‭ ‬(sic‭!) – ‬correct:‭ ‬ochraceus

L74: ‭correct: isolate
L75:‭ correct: ‬producers
L81:‭ correct: ‬extract
L82:‭ ‬300‭ ‬gms‭ – correct: ‬300‭ ‬g
L82,‭ ‬L87,‭ ‬L99:‭ correct: ‬roasted
L86:‭ correct: ‬autoclaving
L91:‭ correct: ‬found

L105:‭ ‬delete one of the duplicated sentences.

L112,‭ ‬L113:‭ ‬oC

Use aw throughout the document‭ (‬lower index‭) ‬L119,‭ ‬L139

Use‭ ‬CO2‭ ‬throughout the document‭ (‬lower index‭) ‬L157,‭ ‬L503

Use‭ ‬o‭‬C throughout the document‭ (‬upper index‭) ‬L158

Latin words such as in vitro,‭ ‬in situ,‭ ‬abbreviation of versus‭ (‬vs.‭) ‬should be italic throughout the document
L181,‭ ‬L182
,‭ ‬and also the name of the moulds‭ (‬L149,‭ ‬L210,‭ ‬L217,‭ ‬L223,‭ ‬L289,‭ ‬L404‭).

L143:‭ ‬water was added

What is ITAL‭ ‬14‭ ‬strain‭? ‬It is just mentioned at L150.

L160:‭ ‬delete‭ ‬:‭ ‬Mycotoxin extraction and analysis

L194,‭ ‬L202:‭ ‬Simonov‭ (‬sic‭!) - ‬The correct name of the test used for checking the normality of the data is: Smirnov

What type of post-hoc test was used in case of parametric data during the ANOVA‭?
(The authors mentioned the test‭ ‬/Kruskal-Wallis/‭ ‬just for the non-parametric‭ ‬data.‭)

‭L203 use this form: non-parametric

L206:‭ ‬use:‭ ‬was set at P‭<‬0.05

L209:‭ correct: ‬media

L212:‭ correct: ‬Overall
L215:‭ correct: ‬very
L221:‭ correct: ‬produced

L221,‭ ‬L453:‭ ‬oC

Fig.‭ ‬1.‭ ‬b‭ – correct: ‬extract

L236:‭ ‬aw

Table‭ ‬1:‭ ‬use‭ ‬±â€­ ‬instead of‭ ‬+

The key for the LOQ abbreviation is false‭ (‬not detection but quantification‭)‬.

L262:‭ ‬At‭ ‬35‭ ‬oC more OTA was produced‭…‬..

L263:‭ correct: ‬coffee

L268:‭ correct: ‬westerdijkiae

L285:‭ ‬delete LOQ and ND

L311:‭ ‬delete the sentence:‭ „‬Please note‭…” ‬as the ranges of Z-axis in Fig.‭ ‬4.‭ ‬a,b are the same.

Move Temperature‭ ‬30oC from L315‭ ‬under L316‭ ‬and above the L318.

Use A.‭ ‬instead of Asp.‭ ‬and correct‭ ‬westerdikiae‭ ‬(sic‭!) ‬to‭ ‬westerdijkiae

The produced OTA concentration on Fig.‭ ‬3‭ (‬a‭) ‬seems to be much higher than it is written in L335‭ (‬43.1‭ ‬ng g-1‭)‬.

‭ L349: correct: stimulation by

L353:‭ ‬delete‭ ‬A.‭ ‬westerdijkiae

In Table‭ ‬5‭ ‬what does the bold numbers mean‭? ‬The significant differences‭? ‬Why is‭ ‬4598.9‭ (‬at L368‭) ‬not bold‭? ‬Where are the SD or SEM values from this table‭?

L375:‭ ‬What does the‭ * ‬mean‭?

L387:‭ ‬what is YES media‭? ‬– this abbreviation is mentioned for the first time [Yeast Extract media]

L408:‭ correct: ‬interacting

L448:‭ ‬use‭ ‬A.‭ ‬westerdijkiae

L452:‭ ‬what is‭ ‬ERH‭? – ‬this abbreviation is mentioned for the first time
[Equilibrium Relative Humidity]

L484:‭ ‬Paterson et al.,‭ ‬2014‭ ‬– they‭ ‬are not mentioned in References

L481:‭ ‬delete:‭ „‬These‭”

L493:‭ correct: ‬increased
L496:‭ correct: ‬coffee

L499:‭ correct: ‬on

In References correction the style according to the guide for authors is necessary. [eg. 1, 3, 4] [13, 16, 17: &, ; , and]

L574: correct: under controlled
L599: correct: CO2

Author Response

See attached file please.
